# A New Species of *Cyrtodactylus tibetanus* Group (Reptilia: Squamata: Gekkonidae) from Xizang Autonomous Region, China [note 1]

**DOI:** 10.3390/ani14162384

**Published:** 2024-08-17

**Authors:** Shun Ma, Sheng-Chao Shi, Tian-Yu Qian, Lu-Lu Sui, Bin Wang, Jian-Ping Jiang

**Affiliations:** 1Chengdu Institute of Biology, Chinese Academy of Sciences, Chengdu 610041, China; mashun21@mails.ucas.ac.cn (S.M.); biomen@foxmail.com (S.-C.S.); qianty@cib.ac.cn (T.-Y.Q.); suill@cib.ac.cn (L.-L.S.); wangbin@cib.ac.cn (B.W.); 2College of Life Science, University of Chinese Academy of Sciences, Beijing 100049, China; 3Hubei Engineering Research Center for Protection and Utilization of Special Biological Resources in the Hanjiang River Basin, School of Life Science, Jianghan University, Wuhan 430056, China

**Keywords:** *Cyrtodactylus laevis* **sp. nov.**, taxonomy, molecular phylogeny, morphological characters, Yarlung Zangbo River

## Abstract

**Simple Summary:**

More than 350 species (32 species groups) of *Cyrtodactylus* Gray, 1827, are reported currently, with a wide distribution from Himalaya, South Asia, Southeast Asia, to Melanesia. *Cyrtodactylus tibetanus* group, as the earliest *Cyrtodactylus* diverging clade, are endemic to the Yarlung–Tsangpo River Basin of Tibetan Plateau, only containing two species: *C. tibetanus* and *C. zhaoermii*. However, two newly collected *Cyrtodactylus* specimens from the eastern region of Lang County, Linzhi City, Xizang Autonomous Region, China, were found to be an unnamed taxon of this group based upon the molecular phylogeny, which was conducted using a concatenation data matrix (the partial 16S ribosomal RNA gene (*16S*), the partial cytochrome c oxidase subunit 1 gene (*CO1*), and the partial NADH dehydrogenase subunit 2 gene (*ND2*)) and morphological comparisons, as the new species could be easily identified from other *C. tibetanus* group species through the absence of the tubercles on dorsum, tail segments, and the tubercles on tails. The new species is currently only known from the type locality with its extremely small populations, with great demand for future investigations to figure out its distribution range, population status, natural history, and mechanisms so that this new species can coexist with *Altiphylax medogense*.

**Abstract:**

A new *Cyrtodactylus* species, *C. laevis*
**sp. nov.**, from the dry-hot valleys near the Yarlung Zangbo River in Re Village, Jindong Countryside, Lang County, Linzhi City, Xizang Autonomous Region, China, is described herein based upon the integrative taxonomic results combining molecular phylogenetic systematics and morphological characteristic comparisons. Our molecular phylogeny was inferred by combining three mitochondrial gene fragments (*16S*/*CO1*/*ND2*), and it indicated a distinct differentiation between the new species and *C. tibetanus* species complex, with obvious genetic distances (*16S* 9.9–11.8%/*CO1* 16.5–18.2%/*ND2* 16.6–18.5%) detected, supporting its validity. Morphologically, the new species can be easily distinguished from its congers by the following characters: (1) medium size (SVL 48.58–50.92 mm), (2) tubercles on dorsum sparse, (3) tail segments absent and tubercles on tails absent, (4) supralabials 10–12 and infralabials 8–10, (5) interorbital scales between anterior corners of the eyes 28–32, (6) scale rows at midbody 96–98, (7) ventral scales between mental and cloacal slit 145–153, (8) ventral scale rows 41–45, and (9) 4 to 5 white–yellow transverse bands with brown dots and black merges between the nape and sacrum. The description of *C. laevis* **sp. nov.** increased the total species number of *C. tibetanus* group to three, and the total *Cyrtodactylus* species number in Xizang to six and in China to eleven. The new species is currently only known from the type locality with its extremely small populations and needs future surveys to reveal its distribution range, population status, natural history, and mechanisms so that the new species can coexist with *Altiphylax medogense*.

## 1. Introduction

Xizang is famous for its complex topography, providing suitable habitats for different amphibian or reptile species. A total of 60 amphibian and 79 reptile species are recorded in this area [1], with a series of new species or supplementary records reported afterwards [2,3,4,5,6,7,8,9,10,11,12,13,14,15,16,17,18,19,20,21,22,23,24,25,26,27,28,29,30,31,32,33,34]. Although a huge number of herpetological surveys have been reported, most studies are limited to southeastern Xizang and lack exploration of the Xizang Plateau.

The bent-toed geckos, *Cyrtodactylus*, possess a huge distribution region, from Himalaya, South Asia, Southeast Asia, to Melanesia, containing well over 350 species, which can be assigned to 32 species groups [35,36]. Among them, ten species from three species groups are recorded from China, respectively: (1) *C. tibetanus* group, containing two species: *C. tibetanus* (Boulenger, 1905) and *C. zhaoermii* Shi and Zhao, 2010; (2) *C. khasiensis* group, containing four species: *C. arunachalensis* Mirza, Bhosale, Ansari, Phansalkar, Sawant, Gowete and Patel, 2021, *C. cayuensis* Li, 2007, *C. dianxiensis* Liu and Rao, 2021, and *C. kamengensis* Mirza, Bhosale, Thackeray, Phansalkar, Sawant, Gowande and Patel, 2022; (3) *C. chauquangensis* group, containing four species: *C. caixitaoi* Liu, Rao, Hou, Wang and Ananjeva, 2023, *C. gulinqingensis* Liu, Li, Hou, Orlov and Ananjeva 2021, *C. menglianensis* Liu and Rao, 2022, and *C. zhenkangensis* Liu and Rao, 2021.

Five species of *Cyrtodactylus* are found in Xizang currently: *C. tibetanus*, *C. zhaoermii*, *C. arunachalensis*, *C. cayuensis*, and *C. kamengensis*, mainly distributed in southeastern Xizang, except species in the *C. tibetanus* group: *C. tibetanus* and *C. zhaoermii*, which are endemic to the Yarlung–Tsangpo River Basin of the Tibetan Plateau, acting as the earliest diverging clade of genus *Cyrtodactylus* [1,37,38], and only comprised by the above-mentioned two species. However, during our field survey along Yarlung Zangbo River from 2023 to 2024, two *Cyrtodactylus* specimens were collected from the east of Lang County, Linzhi City, Xizang Autonomous Region, China (Figure 1). Both morphological and molecular analysis results support them belonging to an unnamed taxon of *C. tibetanus* group and, herein, we describe them as a new species.

## 2. Materials and Methods

### 2.1. Specimens’ Preparation

Two specimens collected by hand from Re Village, Jindong Countryside, Lang County, Linzhi City, Xizang Autonomous Region, China (Figure 1), were preserved in 75% ethanol and deposited in Chengdu Institute of Biology, Chinese Academy of Sciences (CIB, CAS). After being humanely euthanized with an injection of 0.7% tricaine methanesulphonate (MS222) solution, their muscle tissue samples were taken and preserved in 95% ethanol for the further molecular experiment. All animal protocols in this study were reviewed and approved by the Animal Ethical and Welfare Committee of Chengdu Institute of Biology, Chinese Academy of Sciences (permit number: 2020-AR-JJP-01).

### 2.2. Molecular Data and Phylogenetic Analysis

Total genomic DNA was extracted, using the Vazyme FastPure Blood/Cell/Tissue/Bacteria DNA Isolation Mini Kit (Vazyme Biotech Co., Ltd., Nanjing, China), from the muscle tissue samples of each specimen. Three mitochondrial gene fragments were concentrated to conduct phylogenetic analysis, respectively, the partial 16S ribosomal RNA gene (*16S*), the partial cytochrome c oxidase subunit 1 gene (*CO1*), and the partial NADH dehydrogenase subunit 2 gene (*ND2*). For polymerase chain reaction (PCR), *16S* was amplified by primers L3975 (5′-CGCCTGTTTACCAAAAACAT-3′) and H4551 (5′-CCGGTCTGAACTCAGATCACGT-3′) [39], *CO1* was amplified by primers Chmf4 (5′-TYTCWACWAAYCAYAAAGAYATCGG-3′) and Chmr4 (5′-ACYTCRGGRTGRCCRAARAATCA-3′) [40], and *ND2* was amplified by primers rMet-3L (5′-ATACCCCGACAATGTTGG-3′) and rAla-1H (5′-GCCTTAGCTTAATTAAAGTG-3′) [41]. The PCR was performed in 25 μL of reactant with the following cycling conditions (respectively, 16S/ND2/CO1): first, an initial denaturing step at 95 °C for 4/4/5 min; then, 35 cycles of denaturing at 95 °C for 40 s, annealing at 53 °C for 35/40/40 s, and extending at 72 °C for 60 s; finally, a final extending step at 72 °C for 10 min. PCR products were then sequenced by Beijing Qingke New Industry Biotechnology Co., Ltd. (Beijing, China).

*Cyrtodactylus cayuensis* was selected as the outgroup [1], and all the sequences were obtained from this study (Table 1). We input the 16S (492 bp)/CO1 (606 bp)/ND2 (832 bp) into MEGA11 [42], aligned by MUSCLE [43]. Then, we calculated the uncorrected pairwise distances (*p*-distance) in MEGA11 for each sequence. Next, we concatenated the three gene fragments together for future phylogeny construction (1930 bp). IQ-TREE 1.6.12 was performed to conduct Maximum Likelihood (ML) analysis [44] under the best-fit model TPM2+F+G4 computed by ModelFinder, referring to the Bayesian Information Criterion (BIC) [45]. Ultrafast bootstrap approximation (UFB) node support was assessed by using 10,000 ultrafast bootstrap replicates, and UFB (%) ≥ 95 was considered significantly supported [46]. The single-branch tests were conducted using the SH-like approximate likelihood ratio test (SH-aLRT) with 1000 replicates, and nodal support (SH, %) ≥ 80 was also considered well supported [47]. The Bayesian Inference (BI) analysis was conducted via MrBayes [48] in PhyloSuite 1.2.3 [49] using a four-chain run calculated for 10 million generations under the best model, GTR+F+I+G4, computed by ModelFinder for MrBayes analysis, sampling every 1000 samples, with the first 25% of samples discarded as burn-in. Finally, the nodal support Bayesian posterior probabilities (BI, %) ≥ 95 were considered significantly supported.

### 2.3. Morphological Characters and Comparisons

Morphological characters’ data were obtained from the two newly collected specimens. The terminology and methods of measurement of characters and scalation counts followed the work of Zhao et al. [50] and Grismer et al. [51]. Bilateral morphological character measurements and scale feature counts were provided as left/right.

The characters were measured to the nearest 0.01 mm using a Deli Caliper (DL92150): (1) snout–vent length (SVL: from tip of snout to anterior margin of cloaca); (2) tail length (TaL: from posterior margin of cloaca to tip of tail); (3) axilla–groin distance (AGD: distance between axilla and groin); (4) head length (HL: maximum head length from tip of snout to posterior margin of auricular opening); (5) head width (HW: maximum head width measured at the angle of the jaws); (6) head height (HH: maximum head height from the top of the head posterior to the eyes to the bottom of the lower jaw); (7) snout length (SL: from snout tip to anterior corner of the eye); (8) eye–ear distance (EED: distance between posterior margin of the eye to posterior margin of ear opening); (9) maximum eye diameter (ED); (10) maximum ear opening diameter (EAD); (11) maximum rostral width (RW); (12) maximum rostral height (RH); (13) maximum mental width (MW); (14) maximum mental length (ML); (15) forelimb length (FlL: length from the base of the palm to the elbow); (16) hindlimb length (HlL: distance from the base of the heel to the knee).

The scalation features were listed as follows: (1) supralabials (SPL: number of scales from commissure of the jaw to the rostral scale); (2) infralabials (IFL: number of scales from commissure of the jaw to the mental scale); (3) interorbitals (IO: number of scales in a line between anterior corners of the eyes); (4) postmentals (PM: scales bordering the mental); (5) dorsal tubercles row at midbody (DTR); (6) paravertebral tubercles (PVT: counted in a single paravertebral row from the level of the forelimb insertions to the level of the hind limb insertion); (7) scales in a line from mental to the front of the cloacal slit (SMC); (8) scale rows at midbody (SR); (9) ventral scales at midbody (V); (10) subdigital lamellae under entire first finger (LF1); (11) subdigital lamellae under entire fourth finger (LF4); (12) subdigital lamellae under entire first toe (LT1); (13) subdigital lamellae under entire fourth toe (LT4); (14) precloacal pores (PP); (15) postcloacal tubercles (PAT).

For other species of the *C. tibetanus* group, morphological feature information was referred to the newly examined specimens of this study (Appendix B and Appendix A).

## 3. Results

### 3.1. Molecular Phylogenetics

The two unnamed *Cyrtodactylus* specimens clustered together (SH 100/UFB 100/BI 100), forming a sister lineage (SH 80/UFB 74/BI 88) to the *C. tibetanus* clade, although the nodal support was moderately weak (Figure 2). However, relatively deep lineage differentiation with further genetic distances and morphological characters confirmed it should be identified as a new taxon.

The intraspecific genetic distances (*16S*/*CO1*/*ND2*) of the two newly collected *Cyrtodactylus* specimens were 0/0/0.6%, and the interspecific genetic distances amongst them with other *C. tibetanus* group species were, respectively, 9.9–11.8%/16.5–18.2%/16.6–18.5% (Table 2), which is obviously higher than the *p*-distances between *C. tibetanus* and *C. zhaoermii* (8.0–9.1%/15.5–16.2%/16.6–17.2%), indicating that these specimens possess distinct genetic divergences from other *C. tibetanus* group species.

### 3.2. Morphological Comparisons

Morphological measurements and scale counts are shown in Table 3 and details can be found in Appendix A. The two specimens could be easily distinguished from other *C. tibetanus* group species by the absence of the tubercles on dorsum, tail segments, and the tubercles on tails (Figure 3). Moreover, the unnamed species differed from *C. tibetanus*, with more scales in a line from mental to the front of the cloacal slit (SMC 145–153 vs. 123–139) and more ventral scales at the midbody (V 41–45 vs. 31–41), and differed from *C. zhaoermii* by having more supralabials (SPL 10–12 VS. 8), more interorbitals (IO 28–32 VD. 22–24), more scales in a line from mental to the front of the cloacal slit (SMC 145–153 vs. 136–144), more scale rows at the midbody (SR 96–98 vs. 81–89), and more ventral scales at the midbody (V 41–45 vs. 29–32; Table 4).

Therefore, combining the results of molecular phylogenetic and morphological comparisons, the two specimens represent a new species, which we describe herein.

#### Taxonomy

*Cyrtodactylus laevis* **sp. nov.** Ma, Wang and Jiang, 2024


https://zoobank.org/pub:DA68D3DC-36DC-402F-AE7A-21CE5C01BFDC


Figure 3A, Figure 4 and Figure 5.

Holotype. CIB 121654, an adult female, collected on 16 July 2023 (93.3457° E, 29.0442° N; ca. 3060 m a.s.l.), from Re Village, Jindong Countryside, Lang County, Linzhi City, Xizang Autonomous Region, China, by Shun Ma, Lu-Lu Sui, Fei-Rong Ji.

Paratype. CIB 121655, an adult female, collected on 29 May 2024 by Shun Ma and Tian-Yu Qian, with the same collection location as the holotype.

Etymology. This specific epithet refers to its tubercles-sparse skin surface. We suggest “Smooth Bent-Toed Geckos” as its English common name and “光滑裸趾虎” (Guāng Huá Luŏ Zhĭ Hŭ) as its Chinese common name.

Diagnosis. (1) Medium size (SVL 48.58–50.92 mm); (2) tubercles on dorsum are sparse; (3) tail segments are absent and tubercles on tails are absent; (4) supralabials 10–12 and infralabials 8–10; (5) interorbital scales between anterior corners of the eyes 28–32; (6) scale rows at midbody 96–98; (7) ventral scales between mental and cloacal slit 145–153; (8) ventral scale rows 41–45; (9) four to five white–yellow transverse bands with brown dots and black merges between the nape and sacrum.

Description of holotype. An adult female (Figure 4), moderate size, SVL 48.58 mm; body moderately depressed and trunk relatively elongated (AGD/SVL 0.41); tail subequal to the body length.

Head depressed (HH/HL 0.35), length longer than width (HL/HW 0.67), and distinct from neck. Snout rounded at top, elongated (SL/HL 0.41/0.41), and about twice the eye diameter (SL/ED 1.97/1.98). Rostral irregular polygon, nearly twice wider than high (RW/RH 1.99) and wider than mental (RW/MW 1.20). Rostral groove present. Rostral in contact with nostril, first supralabial and nasorostral. Nares oval, touching rostral, first supralabial three nasals (nasorostral, supranasal, and postnasal), and two small internasals. Snout region medially concave and preorbital region concave. Eye large (ED/HL 0.21/0.21), and pupil vertical with crenulated origin-red margins. Interorbital scales between anterior corners of eyes, 32. Ear opening narrow, oval, obliquely oriented, approximately a half-length of the eye diameter (EAD/ED 0.50/0.49). Mental pentagon, with width much larger than length (MW/ML 1.54). Two enlarged postmentals, hexagonal, twice as long as wide. Postmentals in contact with mental and first infralabials anteriorly and a series of gular scales posteriorly. Supralabials, 12/10; infralabials, 9/8. Tubercles absent on dorsal head, and granulars on the anterior dorsal head part larger than those on the posterior part.

Dorsal scales on body, smooth, round, oval, or granular. Dorsal tubercles sparse, and ventrolateral fold absent. Ventrals distinctly larger than dorsal scales, smooth, imbricate, and largest in the middle of the belly. Ventral scale rows at midbody, 45; scale rows around midbody, 98; ventral scales in a row between mental and cloacal slit, 153. Precloacal scales enlarged, extending to thighs.

Forelimbs and hindlimbs well developed, moderately long, and slender (FlL/SVL 0.39; HlL/SVL 0.44/0.45). Tubercles on fore and hind limbs absent. Forearm and tibia moderately long, and forearm shorter than tibia. No webbing on fingers and toes. Subdigital lamellaes on the first finger, 7/6; on the fourth finger, 17/16; on the first toe, 7/8; on the fourth toe, 21/22. Relative length of fingers: IV > III > V > II > I; relative length of toes: IV > III > V > II > I.

Tail oval in section, swollen at base, gradually tapering, and no segments. Postcloacal tubercle, 2/2, obviously large on tail base side. Dorsal scales, small, flat, and smooth, without tubercles. Ventral scales slightly larger than dorsal, smooth, and imbricated, with enlarged subcaudal plates arranging into a longitudinal row formed about 1/6 TaL away from the cloaca.

Coloration of holotype in life. Head dorsum is white–yellow with irregular brown line or dot patterns, and supraorbital regions are black–green, and pink–brown in the preorbital region, comprising a “U” pattern between the two eyes at the nape. Body is pink–brown, with four white–yellow transverse bands, with brown dots and black merges between the nape and sacrum, and a white–yellow dot pattern with black merges at the medium of the 1st and 2nd bands. Dorsal surfaces of limbs are brown–yellow, mottled with small brown patterns. Dorsal tail is pink–brown, ornamented with eight white–yellow transverse bands with brown dots and black merges. Ventral skin is pink–white, mosaiced with small taupe pigments on each scale.

Coloration of holotype in preservative. The specimen’s coloration totally faded. Head dorsum was white–brown with irregular brown line or dot patterns, with the vertex region pale–white. Body was white–brown, with four pale-white transverse bands with light-brown dots and black merges between the nape and sacrum, and a pale-white dot pattern with black merges at the medium of the 1st and 2nd bands. Dorsal surfaces of limbs were also white–brown, mottled with small light-brown patterns. Dorsal tail was milky-white, and the eight white–yellow transverse bands seriously faded, with the merges also greatly faded, leaving only light-brown patterns. Ventral skin was light-brown, mosaiced with small light-brown pigments on each scale.

Variation. The paratype presented a similar morphological pattern as the holotype, and the morphological measurements and scale counts are provided in Table 3, presenting the morphological variations. It had a white–yellow regeneration tail without transverse bands and a white–yellow transverse band with brown dots and black merges at the same position as the white–yellow dot pattern of the holotype (Figure 5).

Distribution and habitats. This new species is only known to be distributed in Re Village, Jindong Countryside, Lang County, Linzhi City, Xizang Autonomous Region, China. The elevation is 3065 m a.s.l. We found the specimens under a rock in the dry-hot valleys near the Yarlung Zangbo River on a sunny afternoon (AT 26 °C, RH 42%; Figure 6), so we consider that it is a nocturnal species. Moreover, *Altiphylax medogense* (Zhao and Li, 1987) and *Laudakia sacra* (Smith, 1935) were also found inhabiting the same area.

## 4. Discussion

The Yarlung Zangbo River basin on the Tibet Plateau is famous for its low reptile species diversity, with only six species recorded in the past [1]. The new finding of *Cyrtodactylus laevis* **sp. nov.** shed light on the importance of continuing to explore the species diversity of reptiles along the Yarlung Zangbo River basin, as it has already been 14 years since the discovery of the last new species, *C. zhaoermii*, in this area [52].

We found a huge number of *Altiphylax medogense* at the type locality of the new species, but only two specimens of the new species were caught from 2023 to 2024, suggesting an extremely limited understanding not only of the population status but also the conservation efforts of *C. laevis* **sp. nov.**, which are essential to improve. Long-term monitoring surveys are needed to bring attention to this issue.

It was revealed that *C. tibetanus* populations from Lang County and Lasa City, Xizang Autonomous Region, China (Figure 1), distinctly formed two different lineages, respectively, in this study, and a similar finding was also revealed by both Che et al. [1] and Grismer et al. [35,38]. A moderate lineage divergence with high nodal supports (SH 100/UFB 100/BI 100; Figure 2) and moderate genetic distances (*16S* 2.3–2.8%/*CO1* 5.6–6.1%/*ND2* 5.9–7.5%) were detected between the two populations, Lang County and Qushui County (Appendix A), whose *CO1*/*ND2 p*-distances were higher than some other recognized *Cyrtodactylus* species. For example, *C. caixitaoi* vs. *C. martini* Ngo, Van, and Tri, 2011 (*CO1* 3.3%) [53]; *C. tayhoaensis* Do, Do, Ngo, Ziegler, Ngo, Nguyen, and Pham, 2023 vs. *C. kingsadai* Ziegler, Phung, Le, and Nguyen, 2013 (*CO1* 4%) [54]; *C. vairengtensis* Lalremsanga, Colney, Vabeiryureilai, Malswmdawngliana, Chandrabohra, Biakzuala, Muansanga, Das, and Purkayastha, 2023 vs. *C. aaronbaueri* Purkayastha, Lalralremsanga, Bohra, Biakzuala, Decemson, Muansanga, Vabeiryureilal, and Rathee, 2021 (*ND2* 3.8–4.2%) [55]. This molecularly confirmed the existence of interspecific differentiation among the two groups. As for morphology, the populations of *C. tibetanus* from Lang County could be identified from the Qushui County population (type locality) according to the different tubercle status (slight vs. medium), especially on tails (slight vs. medium), and relatively weak tail segments (Figure 3B,C), pointing out that the population from Lang County probably represent a cryptic species. Che et al. [1] reported a wide distribution of *C. tibetanus* inhabiting along the Yarlung Zangbo River, indicating that in the future, a wider survey in this area to obtain more genetic information from different populations and conducting phylogenomic analysis to infer the evolutionary pattern and whether there exists any gene flow between these nature populations are required, in order to finally ensure if the *C. tibetanus* populations from Lang County should be regarded as a new taxon and where their species border is.

## 5. Conclusions

Combining the molecular phylogenetic and morphological analysis results, a new *Cyrtodactylus* species, *C. laevis*
**sp. nov.**, from Re Village, Jindong Countryside, Lang County, Linzhi City, Xizang Autonomous Region, China, was described herein, bringing the total species number of the *C. tibetanus* group to three, and the total *Cyrtodactylus* species number in Xizang to six and in China to eleven. *C. laevis* **sp. nov.** is distinct from its congers by the absence of the tubercles on dorsum, tail segments, and the tubercles on tails. The new species is currently only known from the type locality with its extremely small populations, and there is a great need for future investigations to figure out its distribution range, population status, natural history, and mechanisms so that the new species can coexist with *Altiphylax medogense*.

## Figures and Tables

**Figure 1 animals-14-02384-f001:**
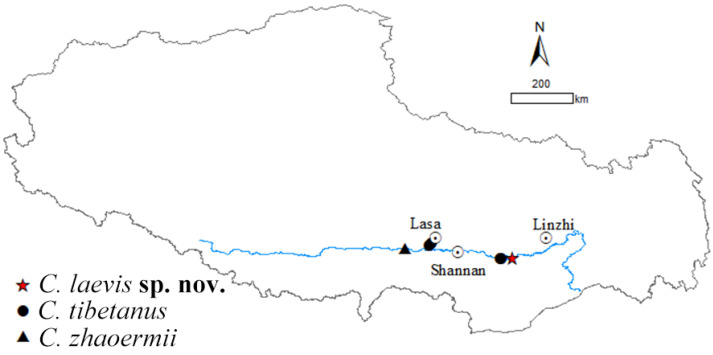
Distribution of *Cyrtodactylus tibetanus* group.

**Figure 2 animals-14-02384-f002:**
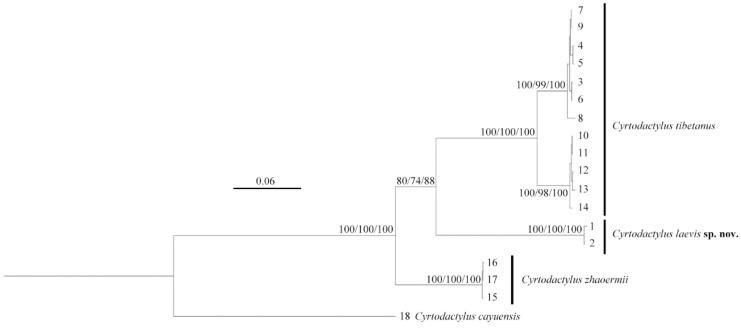
Phylogenetic topology based on three mitochondrial (*16S*/*CO1*/*ND2*) fragments of the *Cyrtodactylus tibetanus* group. The nodes supporting values on branches are presented as the SH-like approximate likelihood ratio test/ultrafast bootstrap approximation/Bayesian posterior probabilities (SH %/UFB %/BI %). Numbers at the tips of branches correspond to the ID numbers in Table 1.

**Figure 3 animals-14-02384-f003:**
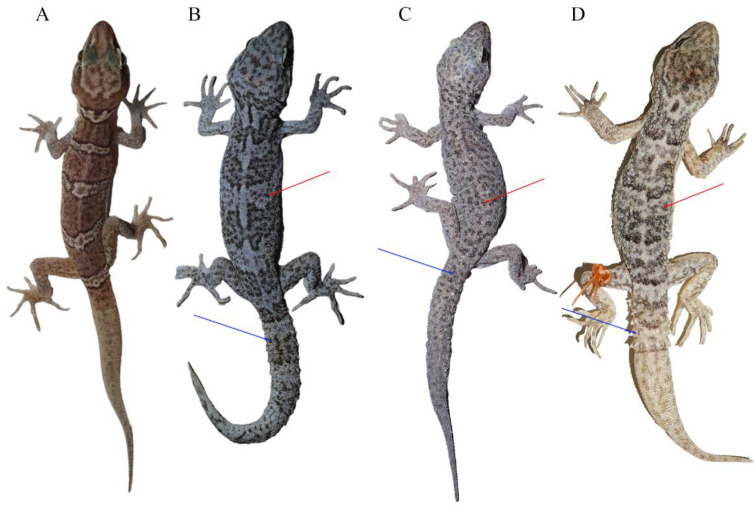
Morphological comparisons of species in the *Cyrtodactylus tibetanus* group: (**A**) *C. laevis* **sp. nov.** (CIB 121655), (**B**) *C. tibetanus* (CIB QZ2021214), (**C**) *C. tibetanus* (CIB QZ2021219), and (**D**) *C. zhaoermii* (CIB XZ2024101). Red arrows indicate the tubercles on dorsum and blue arrows indicate the tail segments and the tubercles on tails. (**A**–**C**) Photos by Sheng-Chao Shi, and (**D**) photo by Shun Ma.

**Figure 4 animals-14-02384-f004:**
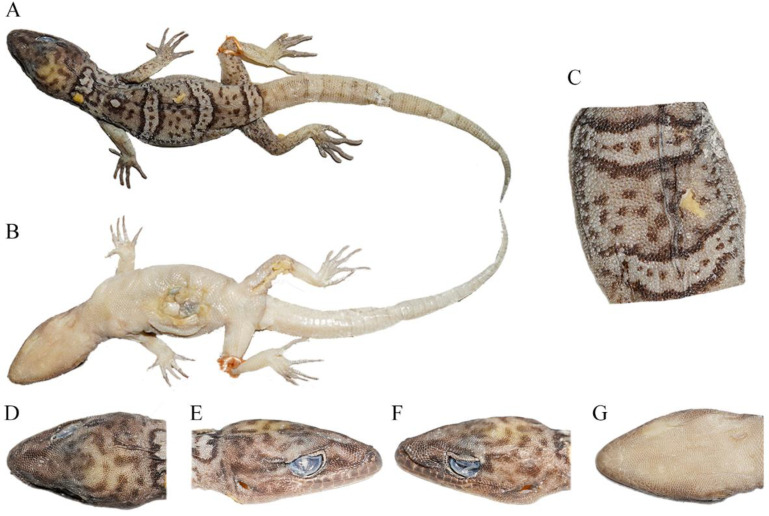
Total views of the *Cyrtodactylus laevis* **sp. nov.** (CIB 121654) holotype in preservative: (**A**) total dorsal view of body, (**B**) total ventral view of body, (**C**) dorsal view of medium dorsum, (**D**) dorsal view of head, (**E**) right-side view of head, (**F**) left-side view of head, and (**G**) ventral-side view of head. Photos by Shun Ma.

**Figure 5 animals-14-02384-f005:**
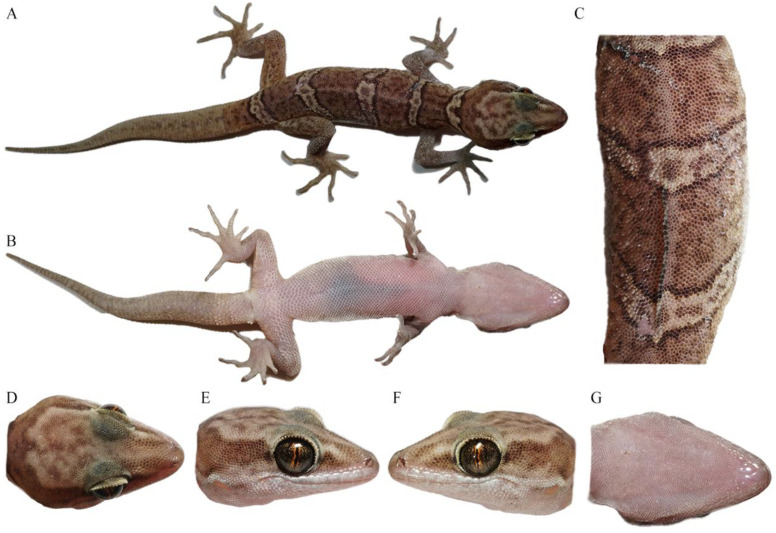
General view of the *Cyrtodactylus laevis* **sp. nov.** (CIB 121655) paratype in life: (**A**) total dorsal view of body, (**B**) total ventral view of body, (**C**) dorsal view of medium dorsum, (**D**) dorsal view of head, (**E**) right-side view of head, (**F**) left-side view of head, and (**G**) ventral-side view of head. Photos by Sheng-Chao Shi.

**Figure 6 animals-14-02384-f006:**
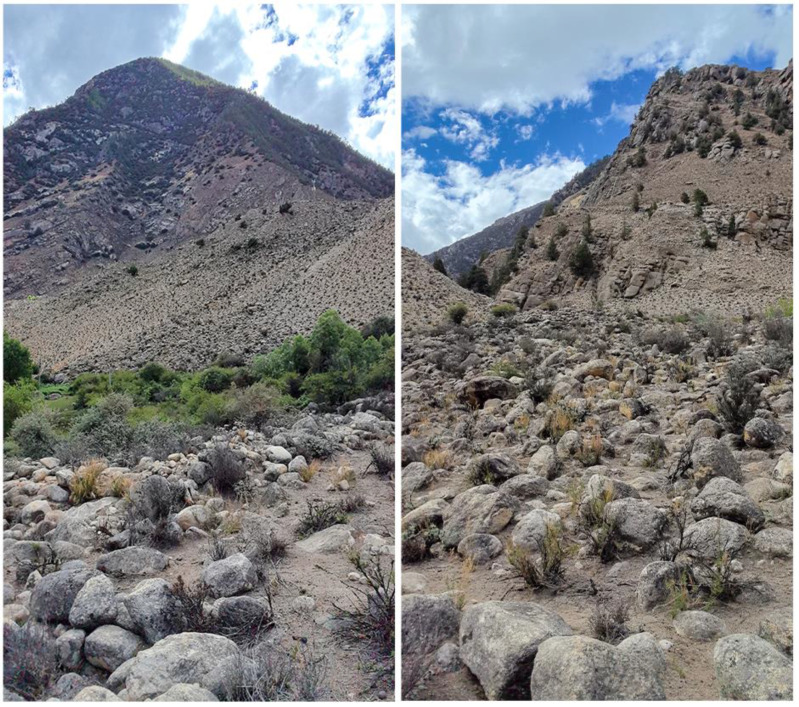
Habitats of *Cyrtodactylus laevis*
**sp. nov.** Photos by Shun Ma.

**Table 1 animals-14-02384-t001:** Information on the *Cyrtodactylus tibetanus* group and *C. cayuensis* sequences used in this study.

No.	Species	Localities	Voucher No.	*16S*/*CO1*/*ND2* GenBank Accession No.	References
1	*C. laevis* **sp. nov.**	Re, Lang, Linzhi, Xizang	CIB 121654	PQ144340/PQ139212/PQ140597	This study
2			CIB 121655	PQ144349/PQ139222/PQ140607	
3	*C. tibetanus*	Nanmu, Qushui, Lasa, Xizang	CIB XZ2024106	PQ144353/PQ139226/PQ140611	
4			CIB XZ2024107	PQ144354/PQ139227/PQ140612	
5			CIB XZ2024108	PQ144355/PQ139228/PQ140613	
6			CIB XZ2024109	PQ144356/PQ139229/PQ140614	
7		Caina, Qushui, Lasa, Xizang	CIB QZ2021212	PQ144342/PQ139214/PQ140599	
8			CIB QZ2021213	PQ144343/PQ139215/PQ140600	
9			CIB QZ2021214	PQ144344/PQ139216/PQ140601	
10		Duiba, Lang, Linzhi, Xizang	CIB QZ386	PQ144345/PQ139217/PQ140602	
11			CIB QZ387	PQ144346/PQ139218/PQ140603	
12			CIB QZ388	PQ144347/PQ139219/PQ140604	
13		Qici, Lang, Linzhi, Xizang	CIB QZ2021217	PQ144348/PQ139220/PQ140605	
14			CIB QZ2021219	——/PQ139221/PQ140606	
15	*C. zhaoermii*	Nimu, Lasa, Xizang	CIB XZ2024099	PQ144350/PQ139223/PQ140608	
16			CIB XZ2024100	PQ144351/PQ139224/PQ140609	
17			CIB XZ2024101	PQ144352/PQ139225/PQ140610	
18	*C. cayuensis*	Xiachayu, Chayu, Linzhi, Xizang	CIB MSJ2023289	PQ144341/PQ139213/PQ140598	

**Table 2 animals-14-02384-t002:** Uncorrected *p*-distances (%) of the *Cyrtodactylus tibetanus* species group based on the mitochondrial *16S*/*CO1*/*ND2* gene sequence fragments.

	1–2 *C. laevis* sp. nov.	3–14 *C. tibetanus*	15–17 *C. zhaoermii*
1–2 *C. laevis* **sp. nov.**	0/0/0.6		
3–14 *C. tibetanus*	11.1–11.8/16.5–18.2/16.6–18.0	0–2.8/0–6.1/0–7.5	
15–17 *C. zhaoermii*	9.9/16.5/17.5–18.5	8.0–9.1/15.5–16.2/16.6–17.2	0/0/0.1–0.2

**Table 3 animals-14-02384-t003:** The measurements and morphological characters of the type series of *Cyrtodactylus laevis* **sp. nov**.

ID	CIB 121654	CIB 121655
Type	Holotype	Paratype
Sex	Female	Female
SVL	48.58 mm	50.92 mm
TaL	51.06 mm	45.82 mm *
AGD	20.12 mm	21.50 mm
HL	14.92 mm	14.66 mm
HW	9.96 mm	9.80 mm
HH	5.22 mm	5.36 mm
SL	6.06/6.10 mm	6.54/6.58 mm
EED	4.20/4.18 mm	3.70/3.68 mm
ED	3.08/3.08 mm	4.26/4.28 mm
EAD	1.54/1.52 mm	1.16/1.20 mm
RW	2.74 mm	2.06 mm
RH	1.38 mm	1.42 mm
MW	2.28 mm	1.98 mm
ML	1.48 mm	1.78 mm
FIL	5.76/5.76 mm	6.86/6.86 mm
HIL	6.54/6.74 mm	8.84/8.88 mm
SPL	12/10	10/10
IFL	9/8	10/9
IO	32	28
PM	2	2
DTR	0	0
PVT	0	0
SMC	153	145
SR	98	96
V	45	41
LF1	7/6	8/8
LF4	17/16	16/15
LT1	7/8	10/10
LT4	21/22	20/20
PP	0	0
PAT	2/2	2/2

* means the length of the regenerated tail.

**Table 4 animals-14-02384-t004:** Morphological comparisons of the *Cyrtodactylus tibetanus* group. Bold represents the differences from *C. laevis* **sp. nov**.

Species	*C. laevis* sp. nov.	*C. tibetanus*	*C. zhaoermii*
SVL	48.58–50.92 mm	49.78–61.16 mm	48.64–57.24 mm
SPL	10–12	7–10	8
IFL	8–10	7–8	8
IO	28–32	23–29	**22–24**
PM	2	2	2
DTR	0	**12–20**	**22–23**
PVT	0	**22–34**	**40–45**
SMC	145–153	**123–139**	**136–144**
SR	96–98	91–107	**81–89**
V	41–45	31–41	**29–32**
LF1	6–8	8–10	8–9
LF4	15–17	15–18	16–19
LT1	8–10	8–10	8–9
LT4	20–22	16–21	19–23
PP	—	4	4
PAT	2	2–4	2–3
Tubercles on dorsum	Absent	**Present, medium, or slight**	**Present, dense**
Tail segments	Absent	**Present or slight only at base**	**Present**
Tubercles on tails	Absent	**Present, medium, or slight**	**Present, dense**

— means data were unavailable.

## Data Availability

The data presented in this study are available on request from the corresponding author.

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
