# Peer review of "A New Species of Cyrtodactylus tibetanus Group (Reptilia: Squamata: Gekkonidae) from Xizang Autonomous Region, China†"

_animals, 2024, doi:10.3390/ani14162384_

Round 1

Reviewer 1 Report

Comments and Suggestions for Authors

Dear authors,

The manuscript generally looks ok. However, there are some issues that need to be corrected prior to its publication. First of all, the new species cannot be published without adding GenBank accession numbers for all the specimens used in the study. I also find it a bit strange that for the phylogenetic study, none of the data from other verified publications (records) have been used. In addition, it would be useful to have a map showing the locality of the species. Also, you do not mention the methodology how the lizards were caught, nor how were they euthanized, and you also do not provide the research permit details.

Below are more detailed comments.

Title- species name has to be in italic

Line 15- at the beginning of the sentence, please spell out full genus name

Line 26- instead of “nature history” write “natural history”

Keywords- delete C. tibetanus and Xizang as both are mentioned in the title

Table 1- provide GenBank accession numbers

Line 122- it is very confusing that you mention five specimens here. From which species did you collect five specimens? You only mention two specimens for the new species; please clarify this; if there are five specimens, please include these data

Line 185, 262- Instead of “regeneration tail” write “regenerated tail”

Figure 2- can you explain different colouration of these species? Here they look very differently coloured so please explain whether these differences are due to preservation, or they look very different in life?

Paragraph 265-270 – instead of “habits” I presume you refer to “habitats”?; please add weather conditions and temperature

Line 270- write “inhabiting” instead of “habiting”

First two paragraphs and the last paragraph of discussion need to be revised as there are many unclear sentences.

Line 300- C. martini has to be in italic

Line 305- instead of “existing” write “existence”

Comments on the Quality of English Language

English is generally ok and only mineor corrections are needed (mostly in the discussion section)

Author Response

Comment: The manuscript generally looks ok. However, there are some issues that need to be corrected prior to its publication. First of all, the new species cannot be published without adding GenBank accession numbers for all the specimens used in the study. I also find it a bit strange that for the phylogenetic study, none of the data from other verified publications (records) have been used. In addition, it would be useful to have a map showing the locality of the species. Also, you do not mention the methodology how the lizards were caught, nor how were they euthanized, and you also do not provide the research permit details.

Response: thanks for your advice, there are limited information of Cyrtodactylus tibetanus group, so we need to sample the specimens by ourselves from the type localities and some other areas and process the lateral molecular laboratory work, thus all the data are original data. We blasted our sequencing results to the records on the GenBank at first to further molecularly help identify the newly-collected specimens although we have identified them morphologically at first. However, those specimens on the GenBank only have one single sequence and vague locality information, so we do not use them. Therefore, our study provided multiple mitochondrial gene fragments of this species group for the first time. We made a map following your suggestion and added the caught method, euthanized method and research permit details.

Comment 1: Title- species name has to be in italic

Response 1: thanks for your advice, we have changed the mode of title-species names to italic.

Comment 2: Line 15- at the beginning of the sentence, please spell out full genus name

Response 2: thanks for your advice, we have spell out full genus name.

Comment 3: Line 26- instead of “nature history” write “natural history”

Response 3: thanks for your advice, we have revised “nature history” to “natural history”.

Comment 4: Keywords- delete C. tibetanus and Xizang as both are mentioned in the title

Response 4: thanks for your advice, we have deleted “C. tibetanus” and “Xizang” in Keywords.

Comment 5: Table 1- provide GenBank accession numbers

Response 5: thanks for your advice, we have provided them.

Comment 6: Line 122- it is very confusing that you mention five specimens here. From which species did you collect five specimens? You only mention two specimens for the new species; please clarify this; if there are five specimens, please include these data

Response 6: thanks for your advice, we only collected two specimens, sorry to make this carelessness.

Comment 7: Line 185, 262- Instead of “regeneration tail” write “regenerated tail”

Response 7: thanks for your advice, we have revised “regeneration tail” to “regenerated tail”.

Comment 8: Figure 2- can you explain different colouration of these species? Here they look very differently coloured so please explain whether these differences are due to preservation, or they look very different in life?

Response 8: they look very different in life, and C. zhaoermii is the similar color when in life.

Comment 9: Paragraph 265-270 – instead of “habits” I presume you refer to “habitats”?; please add weather conditions and temperature

Response 9: thanks for your advice, we have revised “habits” to “habitats” and added weather conditions and temperature.

Comment 10: Line 270- write “inhabiting” instead of “habiting”

Response 10: thanks for your advice, we have revised “habiting” to “inhabiting”.

Comment 11: First two paragraphs and the last paragraph of discussion need to be revised as there are many unclear sentences.

Response 11: thanks for your advice, we have carefully revised the Discussion section.

Comment 12: Line 300- C. martini has to be in italic

Response 12: thanks for your advice, we have changed it to italic

Comment 13: Line 305- instead of “existing” write “existence”

Response 13: thanks for your advice, we have revised “existing” to “existence”

Reviewer 2 Report

Comments and Suggestions for Authors

The presented study is a high-quality description of a new species in terms of content and form, thus an interesting contribution to the knowledge of the species diversity of geckos of the genus Cyrtodactylus and the herpetofauna of China, including remarkable zoogeographical, ecological and conservation connections and implications. I recommend accepting for publication.

I have only marginal formal comments:

The Keywords does not need to contain the terms "C. tibetanus" and Xizang (they are already in the title of the article).

correct typos - Line 65:                                                                                      Ananjeva, 2023, C. gulinqingensis Liu, Li, Hou, Orlov and Ananjeva, 2021

Author Response

Comment 1: The Keywords does not need to contain the terms "C. tibetanus" and Xizang (they are already in the title of the article).

Response 1: thanks for your advice, we have deleted them.

Comment 2: correct typos - Line 65: Ananjeva, 2023, C. gulinqingensis Liu, Li, Hou, Orlov and Ananjeva, 2021

Response 2: thanks for your advice, we have revised them.

Reviewer 3 Report

Comments and Suggestions for Authors

The ms  contains description of a new species from Xizang Autonomous region, China. This area is especially interesting for the study of biodiversity and this new record add new important knowledge about reptile fauna. I have no serious comments to this ms which should be published in journal Animals. However I recommend to made the duscussion more wide even on the base of existing literature including the papers of the authors on Cyrtodactylus genus and add even short note about the life history of this narrow distributed species.   

Author Response

Comment: I recommend to made the discussion more wide even on the base of existing literature including the papers of the authors on Cyrtodactylus genus and add even short note about the life history of this narrow distributed species.

Response: thanks for your advice, we think that some other correct Cyrtodactylus research is not very closed to this study, and for life history, we added the weather conditions, temperature and humidity, but some other life history information is poorly known.

Round 2

Reviewer 1 Report

Comments and Suggestions for Authors

Dear authors,

The manuscript now seems ok, except for the beginning of the discussion section which is still partially confusing. Below are more specific comments on this part of the manuscript.

 Line 327 – change to ...famous for its low reptile...

Line 329 – change to... importance of exploring...

Line 330 – delete „far away“

Line 331 – change to ... since the discovery of ...

Line 333-335- ... transmitting a signal – please rephrase. Unclear!

Line 336-339 - unclear; please rephrase

Comments on the Quality of English Language

Dear authors,

please correct first two paragraphs of discussion section as they are a bit unclear. The rest of the manuscript seems ok.

Author Response

Comment 1: Line 327 – change to ...famous for its low reptile...

Response 1: thanks for your advice, we have changed “as” to “for”.

Comment 2: Line 329 – change to... importance of exploring...

Response 2: thanks for your advice, we have changed “keeping” to “of”.

Comment 3: Line 330 – delete „far away“

Response 3: thanks for your advice, we have deleted “far away”.

Comment 4: Line 331 – change to ... since the discovery of ...

Response 4: thanks for your advice, we have changed “discovering” to “discovery”.

Comment 5: Line 333-335- ... transmitting a signal – please rephrase. Unclear!

Response 5: thanks for your advice, we have rephrased it.

Comment 6: Line 336-339 - unclear; please rephrase

Response 6: thanks for your advice, we have rephrased it.